# Inhibiting NINJ1-dependent plasma membrane rupture protects against inflammasome-induced blood coagulation and inflammation

Jian Cui[1†], Hua Li[1], Dien Ye[1], Guoying Zhang[2], Yan Zhang[2], Ling Yang[2], Martha MS Sim[3‡], Jeremy P Wood[1,3,4], Yinan Wei[2], Zhenyu Li[2], Congqing Wu[1,5,6]*

[1]Saha Cardiovascular Research Center, College of Medicine, University of Kentucky, Lexington, United States; [2]Department of Pharmaceutical Sciences, Irma Lerma Rangel School of Pharmacy, Texas A&M University, College Station, United States; [3]Department of Molecular and Cellular Biochemistry, University of Kentucky, Lexington, United States; [4]The Gill Heart and Vascular Institute, College of Medicine, University of Kentucky, Lexington, United States; [5]Department of Microbiology, Immunology, and Molecular Genetics, College of Medicine, University of Kentucky, Lexington, United States; [6]Department of Surgery, College of Medicine, University of Kentucky, Lexington, United States

*For correspondence:
cwu3@uky.edu

Present address: [†]Department of Medicine, Genetic Medicine Division, Vanderbilt University Medical Center, Nashville, United States; [‡]Bloodworks Northwest Research Institute, Seattle, United States

## eLife Assessment

The authors aim to elucidate the mechanism by which pyroptosis (through the formation of Gasdermin D (GSDMD) pores in the plasma membrane) contributes to increased release of procoagulant Tissue Factor-containing microvesicles. The data offers **solid** mechanistic insights as to the interplay between pyroptosis and microvesicle release with NINJ1. The study provides **useful** insights into the potential of targeting NINJ1 as a therapeutic strategy.

**Abstract** Systemic blood coagulation accompanies inflammation during severe infections like sepsis and COVID. We previously established a link between coagulopathy and pyroptosis, a vital defense mechanism against infection. During pyroptosis, the formation of gasdermin-D (GSDMD) pores on the plasma membrane leads to the release of tissue factor (TF)-positive microvesicles (MVs) that are procoagulant. Mice lacking GSDMD release fewer of these procoagulant MVs. However, the specific mechanisms coupling the activation of GSDMD to MV release remain unclear. Plasma membrane rupture (PMR) in pyroptosis was recently reported to be actively mediated by the transmembrane protein Ninjurin-1 (NINJ1). Here, we show that NINJ1 promotes procoagulant MV release during pyroptosis. Haploinsufficiency or glycine inhibition of NINJ1 limited the release of procoagulant MVs and inflammatory cytokines, and partially protected against blood coagulation and lethality triggered by bacterial flagellin. Our findings suggest a crucial role for NINJ1-dependent PMR in inflammasome-induced blood coagulation and inflammation.

## Introduction

Inflammasomes are intracellular multiprotein complexes that detect pathogens by sensing pathogen-associated molecular patterns (PAMPs) (*Schroder and Tschopp, 2010*; *Lamkanfi and Dixit, 2014*).

The activation of inflammasome and subsequent pyroptosis are critical in host defense against pathogens (*Miao et al., 2010*; *Zhao et al., 2011*; *von Moltke et al., 2012*). Pyroptosis exposes intracellular pathogens through plasma membrane rupture (PMR) (*Nozaki et al., 2022*). However, overactive inflammasomes, common in conditions like sepsis and COVID (*Yap et al., 2020*; *Wu et al., 2021*; *Vora et al., 2021*; *Sefik et al., 2022*; *Junqueira et al., 2022*), can trigger systemic blood coagulation and tissue and organ dysfunction (*Levi and Ten Cate, 1999*; *Levi et al., 2020*; *Al-Samkari et al., 2020*; *Giannis et al., 2020*; *Bowles et al., 2020*).

Tissue factor (TF) is a membrane protein and the primary initiator of the coagulation cascade, essential for hemostasis to stop bleeding upon injury (*Morrissey et al., 1987*; *Mackman, 2004*). Elevated TF activity is associated with consumption coagulopathy in sepsis and COVID (*Taylor et al., 1991*; *Levi et al., 1994*; *Pawlinski et al., 2010*; *Rosell et al., 2021*). Our previous study has shown that TF-positive MVs released by pyroptotic macrophages trigger systemic coagulation and lethality in experimental sepsis (*Wu et al., 2019*). However, the molecular mechanisms by which pyroptosis releases procoagulant MVs remain elusive. Pyroptotic cells undergo PMR at the late stage when the plasma membrane is beyond repair, where the blebbing and pinching of the plasma membrane releases MVs to remove membrane damaged by gasdermin-D (GSDMD) pores (*Ding et al., 2016*; *Liu et al., 2016*; *de Vasconcelos et al., 2019*; *Babiychuk et al., 2011*; *Rühl et al., 2018*). PMR could accelerate MV release as the entire plasma membrane is subject to being dissolved as MVs. Kayagaki et al. recently reported that Ninjurin-1 (NINJ1) oligomerization activates PMR, a process formerly thought to be passive, during lytic cell death like necrosis and pyroptosis (*Kayagaki et al., 2021*). However, the role of NINJ1 in releasing procoagulant MVs and initiating systemic coagulation after inflammasome activation remains unknown.

In this study, we show that *Ninj1* haploinsufficiency inhibits the release of TF-positive MVs and provides protection against systemic coagulation. Reduced NINJ1 also limits the release of inflammatory cytokines. We find that a single dose of glycine injection has similar protective effects as they prevent PMR. Our findings underscore the essential role of NINJ1-dependent PMR in inflammasome-induced blood coagulation, thus expanding our understanding of the dual role that inflammasomes play in host defense.

## Results and discussion
### *Ninj1* haploinsufficiency partially protects against flagellin-induced blood coagulation

We aimed to understand the role of NINJ1 on inflammasome-induced blood coagulation, using heterozygous *Ninj1*[+/-] mice with decreased NINJ1 protein abundance (*Figure 1—figure supplement 1*). This approach was justified by our observation that total NINJ1 deficiency in the mouse strain used in this study caused the reported frequent occurrence of hydrocephalus (*Kayagaki et al., 2023*).

To induce blood coagulation and lethality, we applied our established protocol that involves injection of mice with purified recombinant bacterial flagellin, fused with the cytosolic translocation domain of anthrax lethal factor (LFn). In the presence of anthrax protein protective agent (PA), this flagellin fusion protein (LFn-Fla) can be transported into the cytoplasm (*Zhao et al., 2011*; *Milne et al., 1995*). Flagellin is a PAMP known to activate the NAIP/NLRC4 inflammasome and trigger pyroptosis (*Miao et al., 2010*; *Zhao et al., 2011*; *Lightfield et al., 2008*; *Kofoed and Vance, 2011*). We confirmed robust inflammasome activation and pyroptosis in mouse primary bone marrow-derived macrophages (BMDMs) upon stimulating with purified Fla (LFn-Fla plus PA) (*Figure 1—figure supplement 2A–B*).

Our earlier work has shown that inflammasome-induced systemic coagulation is marked by prolonged prothrombin time (PT) and increased plasma thrombin-antithrombin (TAT) (*Wu et al., 2019*), frequently observed in patients with disseminated intravascular coagulation and COVID (*Levi and Ten Cate, 1999*; *Levi et al., 2020*; *Al-Samkari et al., 2020*; *Giannis et al., 2020*; *Bowles et al., 2020*; *Iba et al., 2019*). As expected, intravenous injection of Fla activated systemic coagulation in *Ninj1*[+/+] mice, demonstrated by a significant increase of PT and plasma TAT (*Figure 1A and B*). Notably, when challenged with Fla, heterozygous *Ninj1*[+/-] mice showed lower PT and almost complete suppression of the elevation in plasma TAT compared with their wild-type littermates (*Figure 1A and B*). Our results suggest that a single copy deletion of *Ninj1* is adequate for significant inhibition of inflammasome-induced blood coagulation.

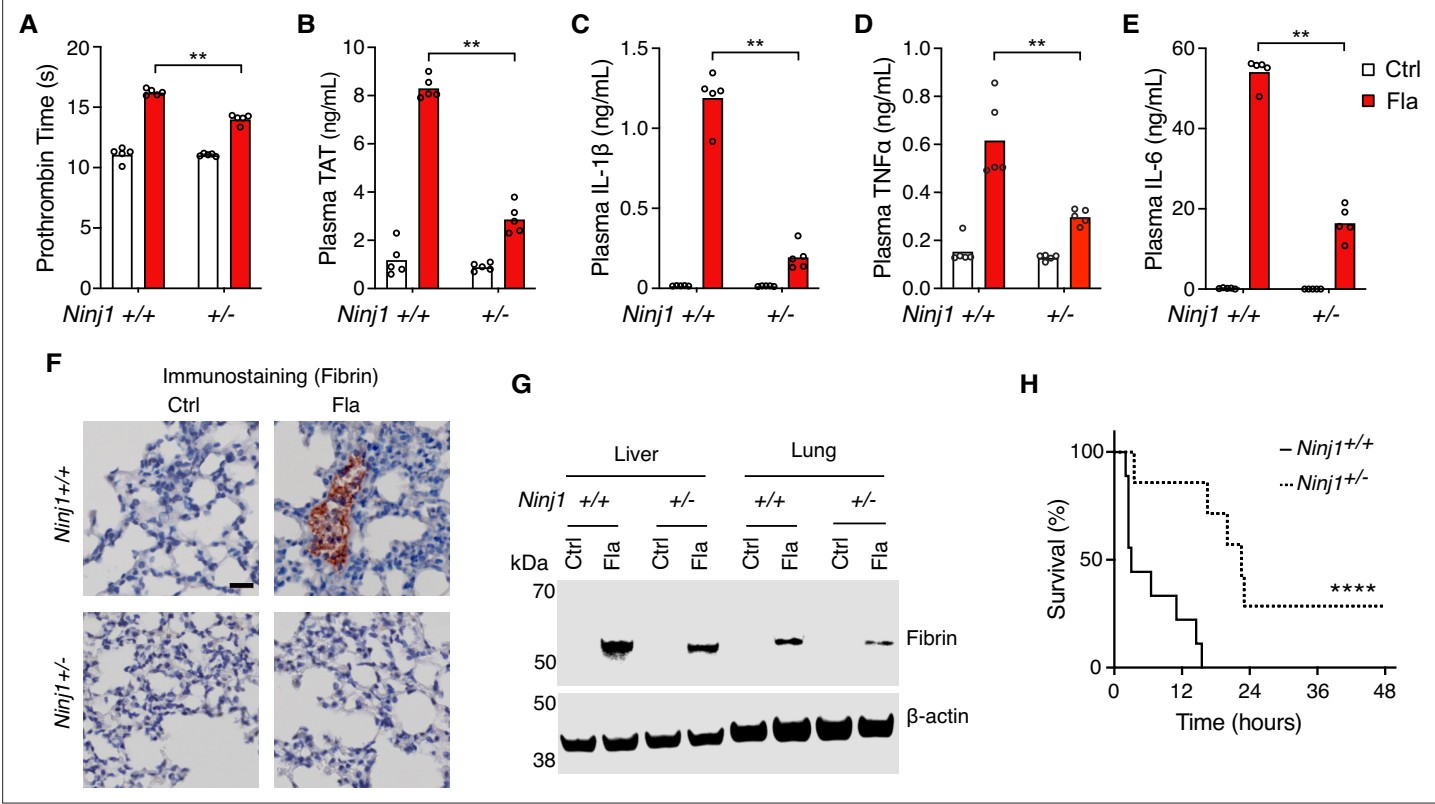

**Figure 1.** NINJ1 is critical for flagellin-induced systemic coagulation, inflammation, and lethality. (**A–E**) Mice were injected intravenously with Ctrl (PBS) or Fla (500 ng LFn-Fla plus 3 µg protective agent [PA]). Blood was collected 90 min after Ctrl or Fla injection. Prothrombin time (**A**), plasma thrombin-antithrombin (TAT) (**B**), and plasma cytokines (**C–E**) were measured. Circles represent individual mice, with bars denoting means. **p<0.01 (two-way ANOVA with Holm-Sidak multiple comparisons). (**F–G**) Mice were injected intravenously with Ctrl or Fla. After 90 min, mice were euthanized and perfused with PBS, and tissues were isolated. (**F**) Lung sections were stained with the anti-fibrin monoclonal antibody (59D8). Scale bar denotes 20 µm. (**G**) Fibrin in the liver and lungs was detected by immunoblot with the anti-fibrin monoclonal antibody (59D8). (**H**) Mice were injected intravenously with a lethal dose of Fla (2.5 µg LFn-Fla plus 6 µg PA). Kaplan-Meier survival plots for mice challenged with Fla are shown. n=7–9. ****p<0.0001 versus WT (log rank test [Mantel-Cox]).

The online version of this article includes the following source data and figure supplement(s) for figure 1:

**Source data 1.** Excel file containing numeric values for panels A-E and H.

**Source data 2.** PDF file containing uncropped western blots with labeling for panel G.

**Source data 3.** Original tiff files of western blots for panel G.

**Figure supplement 1.** NINJ1 protein abundance at baseline in different tissues.

**Figure supplement 1—source data 1.** PDF file containing uncropped western blots with labeling.

**Figure supplement 1—source data 2.** Original tiff files of western blots.

**Figure supplement 2.** Flagellin-induced inflammasome activation and pyroptosis.

**Figure supplement 2—source data 1.** Excel file containing numeric values for panel A.

**Figure supplement 2—source data 2.** PDF file containing uncropped western blots with labeling for panel B.

**Figure supplement 2—source data 3.** Original tiff files of western blots for panel B.

**Figure supplement 3.** Flagellin-induced tissue fibrin deposition (H&E staining) in *Ninj1*$^{+/+}$ and *Ninj1*$^{+/-}$ mice.

We also found that *Ninj1* haploinsufficiency hindered the release of cytokines, as shown by decreased plasma concentrations of cytokines such as IL-1β, TNFα, and IL-6 (***Figure 1C–E***). The decrease in cytokine concentrations could be attributed to limited release of danger-associated molecular patterns (DAMPs) due to less effective PMR in *Ninj1*$^{+/-}$ mice. It's worth noting that GSDMD facilitates IL-1β secretion into cell culture supernatant and NINJ1 deficiency does not affect pyroptosis-induced IL-1β release in vitro (***Kayagaki et al., 2021***; ***Borges et al., 2022***). Given that *Ninj1* haploinsufficiency

did not change GSDMD expression (*Figure 1—figure supplement 1*), there are likely unrecognized mechanisms in pyroptosis-induced IL-1β secretion in vivo.

While our earlier research suggested that cell death, not the release of inflammatory cytokines like IL-1β, is crucial for the over-activation of blood coagulation triggered by inflammasomes (*Wu et al., 2019*), these cytokines are vital for maintaining inflammation. Our findings indicate their release is affected by NINJ1.

### Fibrin deposition and lethality resulted from inflammasome activation depend on NINJ1

Fibrin deposition in tissues can restrict blood supply, potentially resulting in tissue and organ damage and even death (*Levi and Ten Cate, 1999*). Following the Fla challenge, we noted fibrin deposition in the lungs of wild-type mice by immunostaining with the fibrin-specific antibody 59D8 (*Hui et al., 1983*; *Figure 1F*), which was consistent with H&E staining (*Figure 1—figure supplement 3*). However, no signs of fibrin deposition were observed in the lung sections of *Ninj1$^{+/-}$* mice, likely due to the technical limitations of detection on thin (5 μm) tissue sections, compounded by the potential of uneven fibrin distribution. Using immunoblot to examine a homogenized whole lobe of the lung or liver, we detected fibrin deposition in the liver and lungs of *Ninj1$^{+/-}$* mice, but to a much less extent compared with their *Ninj1$^{+/+}$* littermates (*Figure 1G*). Furthermore, *Ninj1* haploinsufficiency partially rescued flagellin-induced lethality (*Figure 1H*). These findings underscore the significant role of NINJ1 protein in inflammasome-triggered fibrin deposition in tissues and lethality.

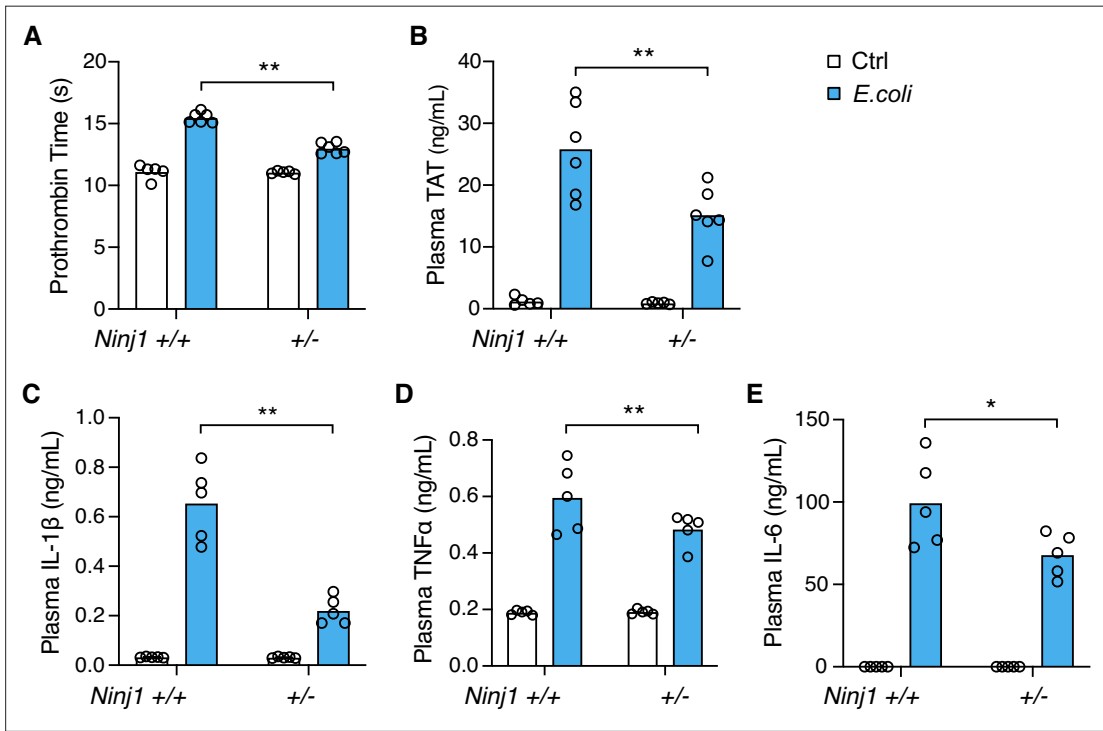

**Figure 2.** *E. coli* infection-induced blood coagulation is limited in *Ninj*1$^{+/-}$ mice. (**A–E**) Mice were injected intraperitoneally with Ctrl (saline) or *E. coli* (2×10$^8$ cfu per mouse). Blood was collected 6 hr afterward. Prothrombin time (**A**), plasma thrombin-antithrombin (TAT) (**B**), and plasma cytokines (**C–E**) were measured. Circles represent individual mice, with bars denoting means. *p<0.05, **p<0.01 (two-way ANOVA with Holm-Sidak multiple comparisons).

The online version of this article includes the following source data for figure 2:

**Source data 1.** Excel file containing numeric values for panel A-E.

## NINJ1 plays a critical role in systemic coagulation in response to *Escherichia coli* infection

Systemic blood coagulation is a common complication of pathogen-triggered sepsis (*Antoniak, 2018*; *Tang et al., 2021*). To investigate the role of NINJ1 in bacterial infection-induced blood coagulation, *Ninj1*[+/-] mice were infected with cultured *E. coli* at a dose of $2\times10^8$ pfu per mouse. Consistently with the findings from the flagellin challenge, *E. coli* infection-induced blood coagulation was limited by *Ninj1* haploinsufficiency. As shown in *Figure 2A and B*, *Ninj1*[+/-] mice exhibited a lower PT and decreased plasma TAT following *E. coli* infection. Furthermore, *Ninj1*[+/-] mice had less cytokine release in response to *E. coli* infection (*Figure 2C–E*). Collectively, these data demonstrate that reduced expression of NINJ1 confers protection against *E. coli* infection-induced blood coagulation. In addition, the reduction in cytokine release in *Ninj1*[+/-] mice supports the role of NINJ1 in modulating the inflammatory response to *E. coli* infection. These findings highlight a potential role of NINJ1 in the pathogenesis of bacterial infection, including blood coagulation and the cytokine storm.

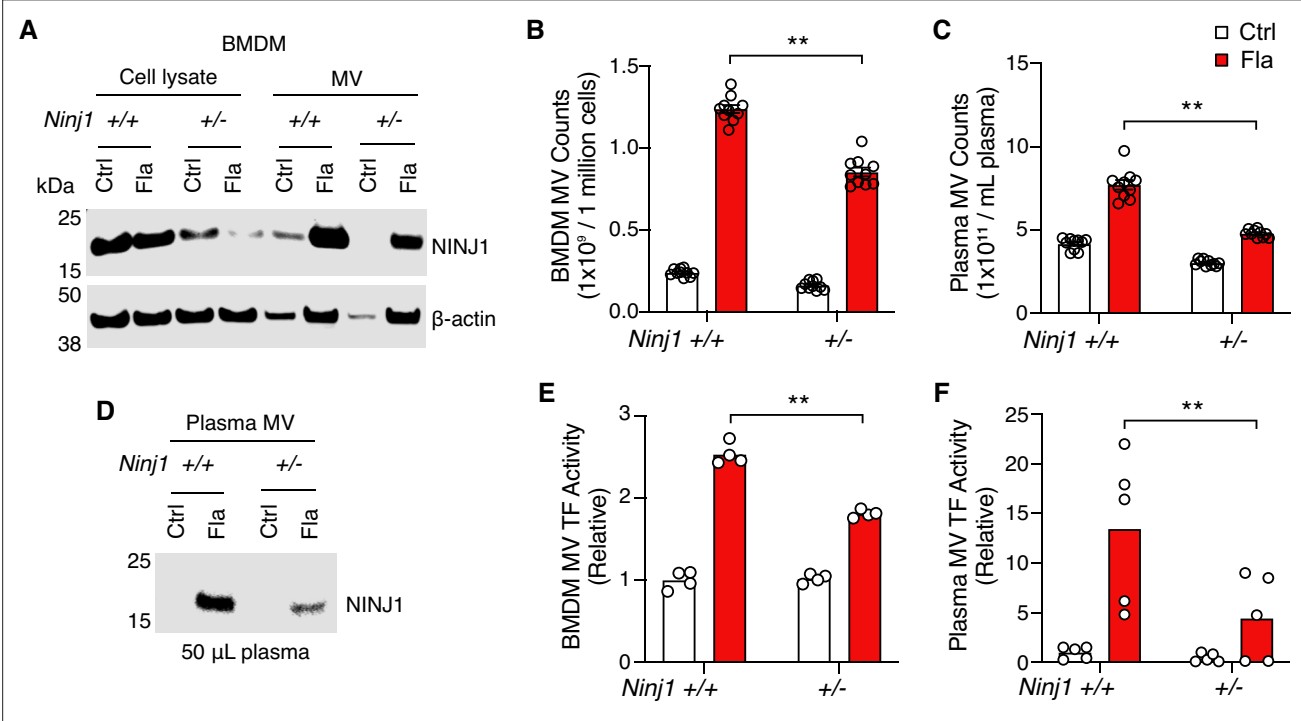

**Figure 3.** Plasma membrane rupture (PMR) promotes the release of procoagulant microvesicles (MVs). (**A–B, E**) Bone marrow-derived macrophages (BMDMs) were incubated with Ctrl (PBS) or Fla (1 µg/mL LFn-Fla plus 1 µg/mL protective agent [PA]). Cell culture supernatant and MVs were collected after 90 min of incubation. (**A**) NINJ1 in cell lysates and MVs was detected by immunoblot. (**B**) BMDM MVs were counted with NanoSight. (**E**) BMDM MV tissue factor (TF) activity. Circles represent individual mouse, with bars denoting means. **p<0.01 (two-way ANOVA with Holm-Sidak multiple comparisons). (**C, D, F**) Mice were injected intravenously with Ctrl (PBS) or Fla (500 ng LFn-Fla plus 3 µg PA). Blood was collected 90 min after Ctrl or Fla injection. (**C**) Plasma MVs were counted with NanoSight. (**D**) NINJ1 in plasma MVs isolated from equal volume of plasma was detected by immunoblot. (**F**) Plasma MV TF activity. Circles represent individual mice, with bars denoting means. **p<0.01 (two-way ANOVA with Holm-Sidak multiple comparisons).

The online version of this article includes the following source data and figure supplement(s) for figure 3:

**Source data 1.** PDF file containing uncropped western blots with labeling for panels A and D.

**Source data 2.** Original tiff files of western blots for panels A and D.

**Source data 3.** Excel file containing numeric values for panels B, C, E, and F.

**Figure supplement 1.** Flagellin-induced pyroptosis and microvesicle (MV) release in *Ninj1*[+/+] and *Ninj1*[+/-] bone marrow-derived macrophages (BMDMs).

**Figure supplement 1—source data 1.** Excel file containing numeric values for panels A-C.

## NINJ1-dependent PMR promotes procoagulant MV release

We have established that macrophage pyroptosis promotes the release of procoagulant MVs (*Wu et al., 2019*). Yet, the cellular processes coupling pyroptosis to MV release remain to be defined. We hypothesize that PMR during pyroptosis is the molecular event responsible for the release of these MVs. To test this hypothesis, we used BMDMs isolated from *Ninj1*[+/+] and *Ninj1*[+/-] littermates. We observed a significant reduction in PMR, indicated by the decreased lactate dehydrogenase (LDH) in the cell culture medium of *Ninj1*[+/-] cells (*Figure 3—figure supplement 1A*), while cell death, measured by ATP levels, remained unaffected (*Figure 3—figure supplement 1B*). This is consistent with previous studies suggesting that NINJ1 mediates PMR but not cell death (*Kayagaki et al., 2021*; *Borges et al., 2022*).

Notably, we detected NINJ1 protein in MVs released from *Ninj1*[+/+] BMDMs (*Figure 3A*). The protein level was decreased in MVs derived from *Ninj1*[+/-] BMDMs (*Figure 3A*), likely due to reduced MV release and/or less NINJ1 protein in *Ninj1*[+/-] cells (*Figure 3B*, *Figure 3—figure supplement 1C*). Consistently, NINJ1 was detected in plasma MVs from *Ninj1*[+/+] mice challenged with Fla (*Figure 3D*). *Ninj1*[+/-] mice also had decreased amount of NINJ1 from MVs, likely due to fewer plasma MVs (*Figure 3C*). The reduced quantity of MVs from both plasma and BMDMs in *Ninj1*[+/-] mice correlated with lower TF activity (*Figure 3E and F*). Our data illustrate that a decline in NINJ1 expression significantly reduces PMR and procoagulant MV release. These findings provide compelling evidence supporting a critical role of NINJ1-dependent PMR in procoagulant MV release. Furthermore, MV shedding removes NINJ1 from the plasma membrane, hinting at a possible role of NINJ1 in cell repair.

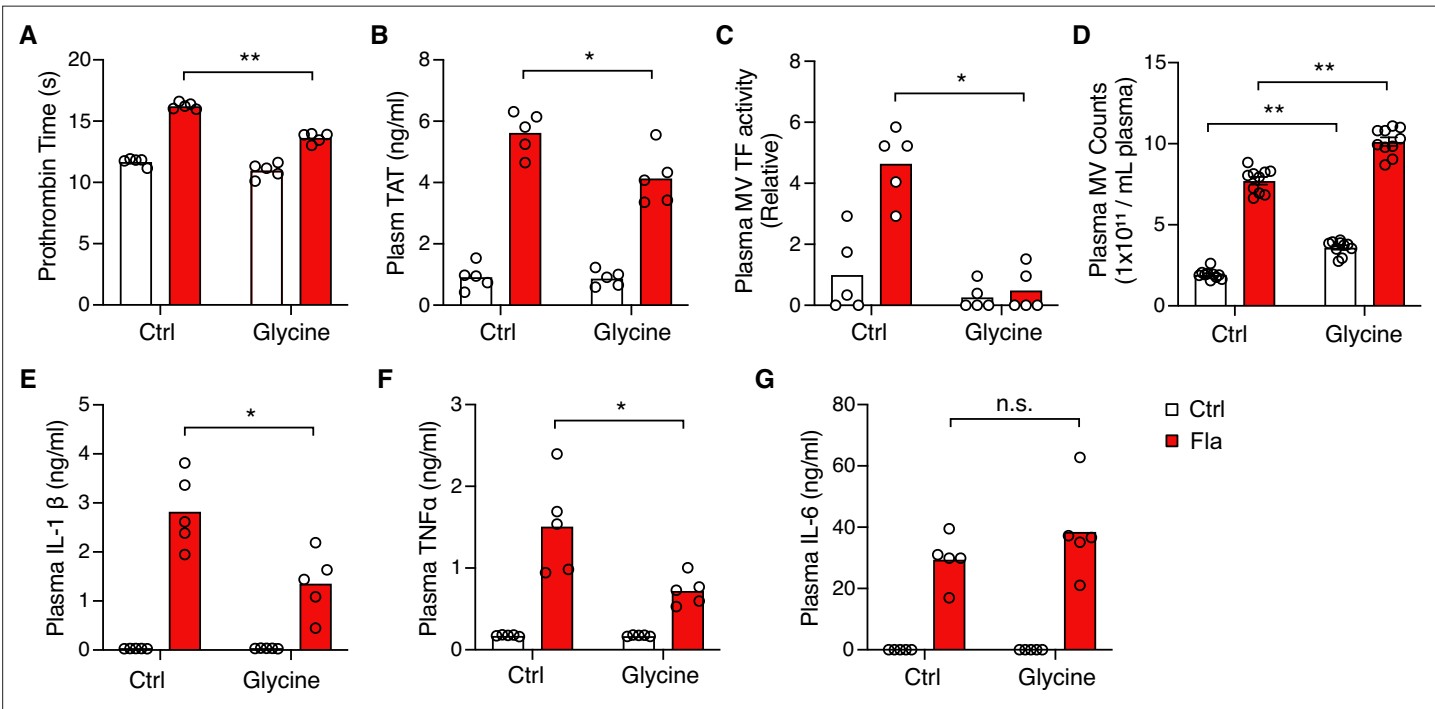

**Figure 4.** Glycine inhibition of NINJ1 blocks pyroptosis-induced blood coagulation. Mice were injected intravenously 50 µL of 0.5 M glycine 2 hr before administrating Ctrl (PBS) or Fla (500 ng LFn-Fla plus 3 µg protective agent [PA]). Blood was collected 90 min after Ctrl or Fla injection. Prothrombin time (A), plasma thrombin-antithrombin (TAT) (B), plasma tissue factor (TF) microvesicle (MV) activity (C), plasma MV counts (D), and plasma cytokines (E–G) were measured. Circles represent individual mice, with bars denoting means. *p<0.05, **p<0.01, n.s. denotes not significant (two-way ANOVA with Holm-Sidak multiple comparisons).

The online version of this article includes the following source data and figure supplement(s) for figure 4:

Source data 1. Excel file containing numeric values for panels A-G.

Figure supplement 1. Glycine treatment inhibits bone marrow-derived macrophage (BMDM) microvesicle (MV) release.

Figure supplement 1—source data 1. Excel file containing numeric values for panels A-G.

# Glycine inhibition of NINJ1 oligomerization reduces pyroptosis-induced blood coagulation

It has long been recognized that glycine buffering protects against PMR during cell death, such as pyroptosis and necrosis (*Weinberg et al., 1987*; *Estacion et al., 2003*). The exact mechanisms underlying the glycine protective effects remained unclear until Borges et al. revealed that glycine inhibits NINJ1 oligomerization and shields cells from rupture (*Borges et al., 2022*). Yet, whether glycine buffering would block the release of PMR-induced TF-positive MVs or limit pyroptosis-induced coagulation in vivo is unknown. To complement our genetic approach, we injected a glycine solution into wild-type mice, prior to Fla injection, aiming to inhibit NINJ1.

Intriguingly, glycine injection led to a significant decrease in PT and plasma TAT (*Figure 4A and B*), demonstrating significant protection against flagellin-induced blood coagulation. Consistently, mice injected with glycine exhibited lower plasma MV TF activity post the Fla challenge (*Figure 4C*). In vitro, glycine buffering on isolated BMDMs protected against PMR, indicated by the reduction in LDH release (*Figure 4—figure supplement 1A*) and decreased MV TF activity (*Figure 4—figure supplement 1B*). However, we observed a discrepancy in MV counts between the in vitro and in vivo studies. Contrary to isolated BMDMs (*Figure 4—figure supplement 1C*), glycine-injected mice had higher MV counts irrespective of the Fla challenge (*Figure 4D*). We speculate that glycine might enhance MV release from cell types other than monocytes and macrophages, independently of inflammasome activation. Lastly, glycine administration reduced the release of proinflammatory cytokines IL-1β and TNFα, consistent with previous research (*Yang et al., 2001*), but not IL-6 (*Figure 4E–G*). In conclusion, the cytoprotective effect of glycine buffering resulted in improved outcomes in inflammasome-induced blood coagulation and inflammation.

In summary, our study employed both genetic and pharmacological approaches to inhibit NINJ1-dependent PMR. Our findings suggest that PMR serves as a pivotal molecular event responsible for the release of procoagulant MVs and the induction of systemic coagulation and inflammation. This novel mechanism sheds light on the potential for developing NINJ1 inhibition as a promising therapeutic strategy for reducing acute inflammation and coagulation-associated tissue and organ damage. In fact, an anti-NINJ1 monoclonal antibody (but not the NINJ1$_{26-37}$ peptide) has been shown to prevent PMR, limiting tissue inflammation in mice (*Kayagaki et al., 2023*). We do, however, recognize the limitations of our main model, as a single injection of bacterial flagellin does not replicate the evolving nature of a polymicrobial infection. Consequently, it cannot offer insights into extended timelines beyond the brief period of flagellin-induced pyroptosis.

## Materials and methods

**Key resources table**

| Reagent type (species) or resource | Designation | Source or reference | Identifiers | Additional information |
|---|---|---|---|---|
| Genetic reagent (*Mus musculus*) | *Ninj1* knockout | Genentech, Inc; *Kayagaki et al., 2021* | | C57Bl/6 strain background |
| Biological sample (*Mus musculus*) | Primary bone marrow-derived macrophages (*Ninj1*$^{+/+}$ BMDMs) | This paper | | Freshly prepared from *Ninj1*$^{+/+}$ male mice at 8–12 weeks of age, described in the Materials and methods section |
| Biological sample (*Mus musculus*) | Primary bone marrow-derived macrophages (*Ninj1*$^{+/-}$ BMDMs) | This paper | | Freshly prepared from *Ninj1*$^{+/-}$ male mice at 8–12 weeks of age, described in the Materials and methods section |
| Strain and strain background (*Escherichia coli*) | *E. coli* strain (EC10) | Dr. Kwang Sik Kim Division of Pediatric Infectious Diseases, Department of Pediatrics, Johns Hopkins University School of Medicine | | |
| Strain and strain background (*Escherichia coli*) | *E. coli* strain ClearColi BL21(DE3) | Lucigen Corporation (now Bioresearch Technologies) | 60810 | |
| Antibody | Anti-mouse NINJ1 (rabbit monoclonal) | Genentech, Inc; *Kayagaki et al., 2021* | Ninj1-rbIgG-25:10363 | Western blot, 1 µg/mL |
| Antibody | Anti-TF antibody (rat monoclonal) | Genentech, Inc | 1H1 | MV TF activity, 100 µg/mL |

*Continued on next page*

*Continued*

| Reagent type (species) or resource | Designation | Source or reference | Identifiers | Additional information |
|---|---|---|---|---|
| Antibody | Anti-β-Actin (mouse monoclonal) | Bio-Rad | MCA5775 | Western blot, 1:1000 |
| Antibody | Anti-Caspase-1 (mouse monoclonal) | Adipogen | AG-20B-0042-C100 | Western blot, 1:1000 |
| Antibody | Anti-GSDMD (rabbit monoclonal) | Abcam | ab219800 | Western blot, 1:1000 |
| Antibody | Anti-IL-1β (rabbit polyclonal) | GeneTex | GTX74034 | Western blot, 1:1000 |
| Antibody | Anti-fibrin (mouse monoclonal) | Gift from Dr. Hartmut Weiler (Medical College of Wisconsin) and Dr. Rodney M Camire (University of Pennsylvania) | 59D8 | Western blot, 1:2000; Immunohistochemistry, 1:250 |
| Commercial assay or kit | CellTiter-Glo Luminescent Cell Viability Assay | Promega | G7572 | |
| Commercial assay or kit | CytoTox 96 Non-Radioactive Cytotoxicity Assay | Promega | G1780 | |
| Commercial assay or kit | Mouse TAT ELISA kit | Abcam | ab137994 | |
| Commercial assay or kit | Mouse IL-1β ELISA kit | Thermo Fisher Scientific | 88-7013A-88 | |
| Commercial assay or kit | Mouse IL-6 ELISA kit | Thermo Fisher Scientific | 88-7064-22 | |
| Commercial assay or kit | Mouse TNFα ELISA kit | Thermo Fisher Scientific | 88-7324-22 | |
| Peptide, recombinant protein | Protective agent (PA) | Zhenyu Li Laboratory Yinan Wei Laboratory | | Described in the Materials and methods section |
| Peptide, recombinant protein | LFn-flagellin | Zhenyu Li Laboratory Yinan Wei Laboratory | | Described in the Materials and methods section |
| Chemical compound and drug | HisPur Ni-NTA resin | Thermo Fisher Scientific | 88222 | |
| Chemical compound and drug | Thromboplastin-D | Pacific Hemostasis | 100357 | |
| Chemical compound and drug | RGR-XaChrom | Enzyme Research Laboratories | 100-03 | |
| Chemical compound and drug | Glycine | Sigma | G7126 | |
| Chemical compound and drug | LPS (*E. coli* O111:B4) | Sigma | L4130 | |
| Chemical compound and drug | T-PER tissue protein extraction reagent | Thermo Fisher Scientific | 78510 | |
| Chemical compound and drug | Cocktail inhibitor | Sigma | P8340 | |
| Software and algorithm | ImageStudio 5.0 | Li-COR | | Western blot image |
| Software and algorithm | Prism 9 | GraphPad Software Inc | | |
| Other | M.O.M. (Mouse on Mouse) ImmPRESS HRP (Peroxidase) Polymer Kit | Vector Laboratories | MP-2400 | Immunohistochemistry |

## Mice

C57BL/6J, *Ninj1*$^{+/+}$, and *Ninj1*$^{+/-}$ mice were housed in the University of Kentucky Animal Care Facility, following institutional and National Institutes of Health guidelines after approval by the Institutional Animal Care and Use Committee.

## Recombinant protein purification

Recombinant proteins were expressed using LPS-free *E. coli* strain ClearColi BL21(DE3) (Lucigen Corporation, Cat#60810) at 37°C for 4 hr with 500 µM IPTG after OD600 reached 0.6–0.8. Bacteria were collected and lysed in 50 mM Tris-HCl and 300 mM NaCl. Proteins containing a His-tag were purified by affinity chromatography using HisPur Ni-NTA resin (Thermo Fisher Scientific, Cat#88222).

Proteins were then eluted with 250 mM imidazole in 50 mM Tris-HCl and 300 mM NaCl, and subsequently dialyzed against PBS to remove imidazole. Proteins concentrations were determined by A280 before sterile filtration.

### In vivo challenges

For flagellin challenge, purified PA and LFn-Fla in PBS were administered via retro-orbital injection. Blood was collected at 90 min post-injection, followed by PBS perfusion. Subsequently, tissue samples were harvested. For *E. coli* challenge, mice were injected intraperitoneally with *E. coli* in saline solution. Blood was collected at 6 hr post-injection. For glycine in vivo experiment, mice were injected intravenously with glycine saline solution 2 hr before flagellin challenge. Blood was collected at 90 min after PBS or flagellin injection.

### BMDM isolation and differentiation

Mouse femur and tibia from one leg were harvested and rinsed in ice-cold PBS, followed by a brief rinse in 70% ethanol for 10–15 s. Both ends of the bones were cut open, and the bone marrow was flushed out using a 10 mL syringe with a 26-gauge needle. The marrow was passed through a 19-gauge needle once to disperse the cells. After filtering through a 70 μm cell strainer, cells were collected by centrifugation at $250 \times g$ for 5 min at 4°C, then suspended in two 150 mm Petri dishes, each containing 25 mL of L-cell conditioned medium (RPMI-1640 supplemented with 10% fetal bovine serum [FBS], 2 mM L-glutamine, 10 mM HEPES, 15% LCM, and penicillin/streptomycin). After 3 days, 15 mL of LCM medium was added to each dish. The cells typically reached full confluency after 5–7 days.

### BMDM culture and stimulation

BMDMs were seeded onto 12-well cell culture plate or 96-well cell culture plate at a density of $1 \times 10^6$ cells/mL of RPMI-1640 medium (Thermo Fisher Scientific, Cat#61870036) containing 10% of FBS (Thermo Fisher Scientific, Cat#A3160502). BMDMs were allowed to settle overnight and refreshed with Opti-MEM (Thermo Fisher Scientific, Cat#51985034) before purified proteins were added. To measure cell cytotoxicity and viability and MVs, supernatant was collected 90 min after stimulation.

### Cell viability and cytotoxicity

Cell viability of BMDMs was determined using CellTiter-Glo Luminescent Cell Viability Assay (Promega, Cat#G7572). Luminescence was recorded as an indicator of ATP levels in metabolically active cells. BMDMs cell cytotoxicity, as measured by LDH in the cell culture supernatant, was determined using CytoTox 96 Non-Radioactive Cytotoxicity Assay (Promega, Cat#G1780) according to the manufacturer's instruction.

### Prothrombin time

Blood was collected from ketamine/xylazine-anesthetized mice by cardiac puncture with a 23-gauge needle attached to a syringe prefilled with 3.8% trisodium citrate as anticoagulant (final ratio at 1:10). Blood was centrifuged at $2,000 \times g$ for 20 min at 4°C to obtain plasma. PT was determined with Thromboplastin-D (Pacific Hemostasis, Cat#100357) in a manual setting according to the manufacturer's instruction, using CHRONO-LOG #367 plastic cuvette.

### Plasma TAT

Plasma was collected as mentioned above in PT. Plasma TAT concentrations were determined using a mouse TAT ELISA kit (Abcam, Cat#ab137994) at 1:50 dilution according to the manufacturer's instruction.

### ELISA

IL-1β, IL-6, and TNFα levels in culture supernatant and plasma were measured with ELISA kits from Thermo Fisher Scientific (#88-7013A-88 for IL-1 β, #88-7064-22 for IL-6, and #88-7324-22 for TNFα) according to the manufacturer's instruction. Plates were read on a Cytation 5 at 450 and 570 nm.

## Tissue preparation and immunohistochemistry

Mice were perfused via both right and left ventricles with PBS. Tissues were collected and embedded in paraffin, then sectioned at 5 µm. Anti-fibrin antibody 59D8 (kindly provided by Dr. Hartmut Weiler at Medical College of Wisconsin and Dr. Rodney M Camire at the University of Pennsylvania) at 4 mg/mL was used to detect fibrin deposition, with M.O.M. (Mouse on Mouse) ImmPRESS HRP (Peroxidase) Polymer Kit (Vector, Cat#MP-2400) from Vector Laboratories according to the manufacturer's instruction for developing positive staining.

## Fibrin extraction for immunoblot

Frozen tissues were homogenized in 20 volumes (mg:µL) of T-PER tissue protein extraction reagent (Thermo Fisher Scientific, Cat#78510) containing cocktail inhibitor (Sigma, Cat#P8340). After centrifugation at 10,000×$g$ for 10 min, supernatant was collected for β-actin detection. Pellets were then homogenized in 3 M urea and vortexed for 2 hr at 37°C. After centrifugation at 14,000×$g$ for 15 min, pellets were suspended in LDS sample buffer and vortexed at 65°C for 30 min and ready for fibrin detection.

## Fluorescent immunoblot

For detection of Caspase-1, IL-1β, GSDMD, and NINJ1 by immunoblot, cells were washed with cold PBS and lysed with LDS sample buffer. Culture supernatant was precipitated with 1/10 volume of 2% sodium deoxycholate and 1/10 volume of 100% trichloroacetic acid, and then dissolved in LDS sample buffer. Caspase-1 was detected using anti-Caspase-1 (Adipogen, Cat#AG-20B-0042-C100) at 1:1000 dilution. IL-1β was detected using anti-IL-1β (GeneTex, Cat#GTX74034) at 1:1000 dilution. GSDMD was detected using anti-GSDMD (Abcam, Cat#ab219800) at 1:1000 dilution. NINJ1 was detected using anti-NINJ1 (kindly provided by Genentech, Cat#Ninj1-rbIgG-25:10363). Tissue fibrin was detected using anti-fibrin (59D8) at 1 mg/mL. Images were acquired with LI-COR Odyssey Imager. Western blot experiments were replicated three times with representative images shown in the figures (*Figures 1G and 3A*).

## Isolation of MVs from mouse plasma

Plasma was collected as mentioned above in PT. Then, 50 µL of mouse plasma was diluted with 1 mL of HBSA (137 mM NaCl, 5.38 mM KCl, 5.55 mM glucose, 10 mM HEPES, 0.1% bovine serum albumin, pH 7.5). MVs were pelleted at 20,000×$g$ for 60 min at 4°C, washed once with 1 mL of HBSA and resuspended in 100 µL HBSA.

## Isolation of MVs from cell culture medium

Cell debris was removed from cell culture supernatant by centrifugation at 1000×$g$ for 10 min. MVs were then pelleted at 20,000×$g$ for 60 min at 4°C and resuspended in HBSA.

## MV counting

To count MVs collected from cell culture supernatant, cell debris was removed by centrifugation at 1000×$g$ for 10 min. MVs in cell culture supernatant or plasma were analyzed directly without further centrifugation. All samples were diluted to optimal conditions (about $10^8$ particles/mL) for analysis in PBS. Video recordings were made for 10 intervals at a length of 30 s each using NanoSight. Nanoparticle tracking analysis was performed to measure the number and size of the plasma and culture supernatant MVs.

## Plasma MV TF activity assay

Plasma (25 µL) was incubated with the 1H1 anti-TF antibody (Genentech) at 100 mg/mL or rat IgG controls for 15 min at room temperature. Next, 25 µL of HBSA containing 10 nM mouse FVIIa, 300 nM human FX, and 10 mM CaCl₂ was added to the samples and incubated for 2 hr at 37°C in a half area 96-well plate. Finally, 12.5 µL of the FXa substrate RGR-XaChrom (4 mM, Enzyme Research Laboratories#100-03) was added and the mixture was incubated at 37°C for 15 min. Absorbance at 405 nm was measured on Cytation 5. The relative TF activity was calculated with absorbance after subtracting the TF-independent activity in the presence of TF blocking antibodies from the total activity in the presence of the IgG control.

## Statistical analysis

Data are represented as individual dots, with bars denoting means. All data represent biological replicates unless stated otherwise. For multiple group with two independent factors, two-way ANOVA with Holm-Sidak multiple comparisons was used for normally distributed variables. A p-value<0.05 was considered significant. Statistical analyses were performed in GraphPad Prism 9.

## Acknowledgements

This work was supported by the National Institutes of Health R00 HL145117 to CW, R01 HL142640 and GM132443 to YW and ZL, R01 HL146744 to ZL, and R00 HL129193 to JPW. Dr. Wendy Katz provided help with tissue paraffin embedding and sectioning and was supported by NIH/NIGMS Institutional Development Award P20GM103527. Dr. Hartmut Weiler at Medical College of Wisconsin and Dr. Rodney M Camire at the University of Pennsylvania provided fibrin antibody. *Ninj1*[+/-] mice and NINJ1 antibody were provided by Genentech, Inc. Dr. Chris Richard provided help with MV counting on NanoSight. Dr. Xu Fu at the University of Kentucky Light Microscopy Core provided help on image acquisition.

## Additional information

### Competing interests

Jeremy P Wood: has an investigator-initiated grant through Pfizer, which is unrelated to this project. The other authors declare that no competing interests exist.

### Funding

| Funder | Grant reference number | Author |
|---|---|---|
| National Heart, Lung, and Blood Institute | R00 HL145117 | Congqing Wu |
| National Heart, Lung, and Blood Institute | R01 HL142640 | Yinan Wei Zhenyu Li |
| National Institute of General Medical Sciences | R01 GM132443 | Yinan Wei Zhenyu Li |
| National Heart, Lung, and Blood Institute | R01 HL146744 | Zhenyu Li |
| National Heart, Lung, and Blood Institute | R00 HL129193 | Jeremy P Wood |

The funders had no role in study design, data collection and interpretation, or the decision to submit the work for publication.

### Author contributions

Jian Cui, Conceptualization, Investigation, Writing – original draft, Writing – review and editing; Hua Li, Dien Ye, Guoying Zhang, Yan Zhang, Ling Yang, Martha MS Sim, Investigation, Writing – review and editing; Jeremy P Wood, Yinan Wei, Zhenyu Li, Writing – review and editing; Congqing Wu, Conceptualization, Supervision, Funding acquisition, Investigation, Writing – original draft, Writing – review and editing

### Author ORCIDs

Congqing Wu ⓘ https://orcid.org/0000-0001-7725-7881

### Ethics

This study was performed in accordance with the recommendations in the Guide for the Care and Use of Laboratory Animals of the National Institutes of Health. All of the animals were housed and handled according to approved Institutional Animal Care and Use Committee (IACUC) protocol (#2021-3751) at the University of Kentucky.

Reviewer #1 (Public review): https://doi.org/10.7554/eLife.91329.3.sa1
Reviewer #2 (Public review): https://doi.org/10.7554/eLife.91329.3.sa2
Author response https://doi.org/10.7554/eLife.91329.3.sa3

## Additional files

### Supplementary files
MDAR checklist

### Data availability
All data generated or analyzed during this study are included in the manuscript and supporting files, which includes the source data for the manuscript figures.

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
