## [Editor Report · eLife Assessment]

The authors aim to elucidate the mechanism by which pyroptosis (through the formation of Gasdermin D (GSDMD) pores in the plasma membrane) contributes to increased release of procoagulant Tissue Factor-containing microvesicles. The data offers **solid** mechanistic insights as to the interplay between pyroptosis and microvesicle release with NINJ1. The study provides **useful** insights into the potential of targeting NINJ1 as a therapeutic strategy.

---

## [Referee Report · Reviewer #1 (Public review)]

The authors demonstrated that NINJ1 promotes TF-positive MV release during pyroptosis and thereby triggers coagulation. Coagulation is one of the risk factors that can cause secondary complications in various inflammatory diseases, making it a highly important therapeutic target in clinical treatment. This paper effectively explains the connection between pyroptosis and MV release with Ninj1, which is a significant strength. It provides valuable insight into the potential of targeting Ninj1 as a therapeutic strategy.

Although the advances in this paper are valuable, several aspects need to be clarified. Some comments are discussed below.

(1) Since it is not Ninj1 directly regulating coagulation but rather the MV released by Ninj1 playing a role, the title should include that. The current title makes it seem like Ninj1 directly regulates inflammation and coagulation. It would be better to revise the title.

(2) Ninj1 is known to be an induced protein that is barely expressed in normal conditions. As you showed in "Fig1G" data, control samples showed no detection of Ninj1. However, in "Figure S1", all tissues (liver, lung, kidney and spleen) expressed Ninj1 protein. If the authors stimulated the mice with fla injection, it should be mentioned in the figure legend.

(3) In "Fig3A", the Ninj1 protein expression was increased in the control of BMDM +/- cell lysate rather than fla stimulation. However, in MV, Ninj1 was not detected at all in +/- control but was only observed with Fla injection. The authors need to provide an explanation for this observation. Additionally, looking at the MV β-actin lane, the band thicknesses appear to be very different between groups. It seems necessary to equalize the protein amounts. If that is difficult, at least between the +/+ and +/- controls.

(4) Since the authors focused Ninj1-dependent microvesicle (MV) release, they need to show MV characterizations (EM, NTA, Western for MV markers, etc...).

(5) To clarify whether Ninj1-dependent MV induces coagulation, the authors need to determine whether platelet aggregation is reduced with isolated +/- MVs compared to +/+ MVs.

(6) Even with the authors well established experiments with haploid mice, it is a critical limitation of this paper. To improve the quality of this paper, the authors should consider confirming the findings using mouse macrophage cell lines, such as generating Ninj1-/- Raw264.7 cell lines, to examine the homozygous effect.

(7) There was a paper reported in 2023 (Zhou, X. et al., NINJ1 Regulates Platelet Activation and PANoptosis in Septic Disseminated Intravascular Coagulation. Int. J. Mol. Sci. 2023) that revealed the relationship between Ninj1 and coagulation. According to this paper, inhibition of Ninj1 in platelets prevents pyroptosis, leading to reduced platelet activation and, consequently, the suppression of thrombosis. How about the activation of platelets in Ninj1 +/- mice? The author should add this paper in the reference section and discuss the platelet functions in their mice.

---

## [Referee Report · Reviewer #2 (Public review)]

Summary:

The authors main goal is to understand the mechanism by which pyroptosis (through the formation of Gasdermin D (GSDMD) pores in the plasma membrane) contributes to increased release of procoagulant Tissue Factor-containing microvesicles (MV). Their previous data demonstrate that GSDMD is critical for the release of MV that contains Tissue Factor (TF), thus making a link between pyroptosis and hypercoagulation. Given the recent identification of NINJ1 being responsible for plasma membrane rupture (Kayagaki et al. Nature 2011), the authors wanted to determine if NINJ1 is responsible for TF-containing MV release. Given the constitutive ninj1 KO mouse leads to partial embryonic lethality, the authors decide to use a heterozygous ninj1 KO mouse (ninj1+/-), and demonstrate that Ninj1 plays a role in release of TF-containing MV.

---

## [Author Response]

The following is the authors’ response to the current reviews.

**Public Reviews:**

**Reviewer #1 (Public Review):**
The authors demonstrated that NINJ1 promotes TF-positive MV release during pyroptosis and thereby triggers coagulation. Coagulation is one of the risk factors that can cause secondary complications in various inflammatory diseases, making it a highly important therapeutic target in clinical treatment. This paper effectively explains the connection between pyroptosis and MV release with Ninj1, which is a significant strength. It provides valuable insight into the potential of targeting Ninj1 as a therapeutic strategy.Although the advances in this paper are valuable, several aspects need to be clarified. Some comments are discussed below.(1) Since it is not Ninj1 directly regulating coagulation but rather the MV released by Ninj1 playing a role, the title should include that. The current title makes it seem like Ninj1 directly regulates inflammation and coagulation. It would be better to revise the title.

Thanks for the thoughtful comments. We show that the release of procoagulant MVs by plasma membrane rupture (PMR) is a critical step in the activation of coagulation. In addition, the release of cytokines and danger molecules by PMR may also contribute to coagulation. In choosing the title, we are trying to emphasize NINJ1-dependent PMR as a common trigger for these biological processes.

(2) Ninj1 is known to be an induced protein that is barely expressed in normal conditions. As you showed in "Fig1G" data, control samples showed no detection of Ninj1. However, in "Figure S1", all tissues (liver, lung, kidney and spleen) expressed Ninj1 protein. If the authors stimulated the mice with fla injection, it should be mentioned in the figure legend.

We respectfully disagree with the comment that “*Ninj1 is known to be an induced protein that is barely expressed in normal conditions*”. NINJ1 protein is abundantly expressed (without induction) in tissues including liver, lung, kidney, and spleen, as shown in Fig S1. Consistently, other groups have shown abundant NINJ1 expression at baseline in tissues and cells such as liver (Kayagaki *et.al.* Nature 2023) and BMDM (Kayagaki *et.al.* Nature 2021; Borges *et.al.* eLife 2022). Fig 1G shows fibrin deposition as an indicator of coagulation, not NINJ1 protein.

(3) In "Fig3A", the Ninj1 protein expression was increased in the control of BMDM +/- cell lysate rather than fla stimulation. However, in MV, Ninj1 was not detected at all in +/- control but was only observed with Fla injection. The authors need to provide an explanation for this observation. Additionally, looking at the MV β-actin lane, the band thicknesses appear to be very different between groups. It seems necessary to equalize the protein amounts. If that is difficult, at least between the +/+ and +/- controls.

Thanks for the valuable comments. In Fla-stimulated Ninj1+/- BMDMs, most of the NINJ1 is released in MVs, therefore, not in the cell lysate, as shown in Fig 3A. The difference in beta-actin band intensity correlated with MV numbers shown in Fig 3B. We ensure consistency by using the same number of cells.

(4) Since the authors focused Ninj1-dependent microvesicle (MV) release, they need to show MV characterizations (EM, NTA, Western for MV markers, etc...).

Thanks for the suggestion. We now add NTA analysis of MV for BMDMs in Fig S4C.

(5) To clarify whether Ninj1-dependent MV induces coagulation, the authors need to determine whether platelet aggregation is reduced with isolated +/- MVs compared to +/+ MVs.

Thanks for the suggestion. We agree that platelet aggregation is closely linked to blood coagulation but would argue that one does not directly cause the other. While it would be interesting to examine whether MVs induce platelet aggregation, we hope the reviewer would agree that the outcome of this experiment would neither significantly support nor challenge our statement that NINJ1-dependent PMR promotes coagulation.

(6) Even with the authors well established experiments with haploid mice, it is a critical limitation of this paper. To improve the quality of this paper, the authors should consider confirming the findings using mouse macrophage cell lines, such as generating Ninj1-/- Raw264.7 cell lines, to examine the homozygous effect.

Thanks for the valuable comments. We acknowledge the limitation of using haploid mice in this study. However, our data provides strong evidence supporting the role of NINJ1-dependent plasma membrane rupture in blood coagulation using primary macrophages.

(7) There was a paper reported in 2023 (Zhou, X. et al., NINJ1 Regulates Platelet Activation and PANoptosis in Septic Disseminated Intravascular Coagulation. Int. J. Mol. Sci. 2023) that revealed the relationship between Ninj1 and coagulation. According to this paper, inhibition of Ninj1 in platelets prevents pyroptosis, leading to reduced platelet activation and, consequently, the suppression of thrombosis. How about the activation of platelets in Ninj1 +/- mice? The author should add this paper in the reference section and discuss the platelet functions in their mice.

Thanks for the valuable comments. We examine PT time, plasma TAT, and tissue fibrin deposition as direct evidence of blood coagulation in this manuscript. We acknowledge that platelets play a key role in thrombosis; however, we hope the reviewer would agree that tissue factor-induced blood coagulation and platelet aggregation are linked yet distinct processes. Therefore, the role of NINJ1 in platelet aggregation falls beyond the scope of this manuscript.

The following is the authors’ response to the original reviews.

**Public Reviews:**

**Reviewer #1 (Public Review):**
Referring to previous research findings, the authors explain the connection between NINJ1 and MVs. Additional experiments and clarifications will strengthen the conclusions of this study.Below are some comments I feel could strengthen the manuscript:(1) The authors mentioned their choice of using heterozygous NINJ1+/- mice on page 4, because of lethality and hydrocephalus. Nonetheless, there is a substantial number of references that use homozygous NINJ1-/- mice. Could there be any other specific reasons for using heterozygous mice in this study?

Thanks for the thoughtful comments. We are aware that a few homozygous NINJ1-/- mouse strains were used in several publications by different groups, including Drs. Kayagaki and Dixit (Genentech), from whom we obtained the heterozygous NINJ1+/- breeders. We do not have experience with the homozygous NINJ1-/- mice used by other groups. It’s reasonable to assume that homozygous NINJ1-/-, if healthy, would have even stronger protection against coagulopathy than heterozygous NINJ1+/-. The only reason for not using homozygous mice in this study is that a majority of our homozygous NINJ1-/- develops hydrocephalus around weaning and these mice are required to be euthanized by the rules of our DLAR facility. Although our homozygous NINJ1-/- mice develop hydrocephalus (the same reported by Drs. Kayagaki and Dixit, PMID: 37196676, PMCID: PMC10307625), heterozygous NINJ1+/- mice remain healthy.

(2) Figure S2 clearly shows the method of pyroptosis induction by flagellin. It is also necessary as a prerequisite for this paper to show the changes in flagellin-induced pyroptosis in heterozygous NINJ1+/- mice.

Thanks for the valuable suggestions. We agree that a plasma LDH measurement as an indicator of pyroptosis *in vivo* would add to the manuscript. Therefore, we have made several attempts to measure plasma LDH in flagellin-challenged WT and NINJ1+/- mice using CytoTox96 Non-Radioactive Cytotoxicity Assay (a Promega kit commonly used for LDH, Promega#G1780). Flagellin-challenged WT and NINJ1+/- mice develops hemolysis, which renders plasma red. Because plasma coloring interferes with the assay, we could not get a meaningful reading to make an accurate comparison. We also tried LHD-Glo Cytotoxicity Assay (Luciferase based, Promega#J2380) with no luck on both plasma and serum. We hope the reviewer would agree that reduced plasma MV count (Fig 3C) would serve as an alternative indictor for reduced pyroptosis.

(3) IL-1ß levels controlled by GSDMD were not affected by NINJ1 expression according to previous studies (Ref 37, 29, Nature volume 618, pages 1065-1071 (2023)). GSDMD also plays an important role in TF release in pyroptosis. Are GSDMD levels not altered in heterozygous NINJ1 +/- mice?

Thanks for raising these great points. It’s been reported that IL-1β secretion in cell culture supernatant were not affected by NINJ1 deficiency or inhibition when BMDMs were stimulated by LPS (Ref 29, 37, now Ref 29, 35) or nigericin (Ref 29). As GSDMD pore has been shown to facilitate the release of mature IL-1β, these *in vitro* observations are reasonable given that NINJ1-mediated PMR is a later event than GSDMD pore-forming. However, we observed that plasma IL-1β (also TNFα and IL-6) in Ninj1+/- mice were significantly lower. There are a few differences in the experimental condition that might contribute to the discrepancy: 1, there was no priming in our *in vivo* experiment, while priming in BMDMs were performed in both *in vitro* observations before stimulating with LPS or nigericin; 2, the flagellin in our study engages different inflammasome than either LPS or nigericin. Priming might change the expression and dynamics of IL-1β. More importantly, there might be unrecognized mechanisms in IL-1β secretion *in vivo*. We now add discussion on this in the main text.

We examined GSDMD protein levels in liver, lung, kidney, and spleen from WT and NINJ1+/- mice by Western blotting. The data is now presented in the updated Fig S1, we did not observe apparent difference in GSDMD expression between the two genotypes.

(4) In Fig 1 F, the authors used a fibrin-specific monoclonal antibody for staining fibrin, but it's not clearly defined. There may be some problem with the quality of antibody or technical issues. Considering this, exploring alternative methods to visualize fibrin might be beneficial. Fibrin is an acidophil material, so attempting H&E staining or Movat's pentachrome staining might help for identify fibrin areas.

Thanks for the valuable suggestions. The fibrin-specific monoclonal antibody in our study is mouse anti-fibrin monoclonal antibody (59D8). This antibody has been shown to bind to fibrin even in the presence of human fibrinogen at the concentration found in plasma [Hui et al. (1983). Science. 222 (4628); 1129-1132]. We apologize that we did not cite the reference in our initial submission. We obtained this antibody from Dr. Hartmut Weiler at Medical College of Wisconsin and Dr. Rodney M. Camire at the University of Pennsylvania, who were acknowledged in our initial submission.

We performed H&E staining on serial sections of the same tissues for Figure 1F. The data is now presented as Fig S3.

**Reviewer #2 (Public Review):**
Summary:The author's main goal is to understand the mechanism by which pyroptosis (through the formation of Gasdermin D (GSDMD) pores in the plasma membrane) contributes to increased release of procoagulant Tissue Factor-containing microvesicles (MV). Their previous data demonstrate that GSDMD is critical for the release of MV that contains Tissue Factor (TF), thus making a link between pyroptosis and hypercoagulation. Given the recent identification of NINJ1 being responsible for plasma membrane rupture (Kayagaki et al. Nature 2011), the authors wanted to determine if NINJ1 is responsible for TF-containing MV release. Given the constitutive ninj1 KO mouse leads to partial embryonic lethality, the authors decided to use a heterozygous ninj1 KO mouse (ninj1+/-). While the data are well controlled, there is limited understanding of the mechanism of action. Also, given that the GSDMD pores have an ~18 nm inner diameter enough to release IL-1β, while larger molecules like LDH (140 kDa) and other DAMPs require plasma membrane rupture (likely mediated by NINJ1), it s not unexpected that large MVs require NINJ1-mediated plasma cell rupture.Strengths:The authors convincingly demonstrate that ninj1 haploinsufficiency leads to decreased prothrombin time, plasma TAT and plasma cytokines 90 minutes post-treatment in mice, which leads to partial protection from lethality.Weaknesses:- In the abstract, the authors say "...cytokines and protected against blood coagulation and lethality triggered by bacterial flagellin". This conclusion is not substantiated by the data, as you still see 70% mortality at 24 hours in the ninj1+/- mice.

Thanks for the thoughtful comments. We corrected the text to “partially protected against blood coagulation and lethality triggered by bacterial flagellin”.

- The previous publication by the authors (Wu et al. Immunity 2019) clearly shows that GSDMDdependent pyroptosis is required for inflammasome-induced coagulation and mouse lethality. However, as it is not possible for the authors to use the homozygous ninj1 KO mouse due to partial embryonic lethality, it becomes challenging to compare these two studies and the contributions of GSDMD vs. NINJ1. Comparing the contributions of GSDMD and NINJ1 in human blood-derived monocytes/macrophages where you can delete both genes and assess their relevant contributions to TF-containing MV release within the same background would be crucial in comparing how much contribution NINJ1 has versus what has been published for GSDMD? This would help support the in vivo findings and further corroborate the proposed conclusions made in this manuscript.

Thanks for the valuable question. We have shown that plasma MV TF activity was reduced in both GSDMD deficient mice (Ref 23) and Ninj1+/- mice (present manuscript). Given that TF is a plasma membrane protein, MV TF most likely comes from ruptured plasma membrane. In flagellin-induced pyroptosis, both GSDMD and NINJ1 deficiency equally blocked LDH release (plasma membrane rupture) in BMDMs (Ref 29). Further, in pyroptosis glycine acts downstream of GSDMD pore formation for its effect against NINJ1 activation (Ref 35). Therefore, GSDMD pore-forming should be upstream of NINJ1 activation in pyroptosis (which may not be the case in other forms of cell death) and there are likely equal effects of GSDMD and NINJ1 on MV release in flagellin-induced pyroptosis. As the reviewer suggested, experiments using human blood-derived monocytes/macrophages will enable a direct comparison to determine the relative contribution. However, this approach presents a few technical difficulties: it’s not easy to manipulate gene expression on primary human monocytes/macrophages (in our experience); variable efficiency in gene manipulation of GSDMD and NINJ1 will complicate the comparison. I hope the reviewer would agree that a direct comparison between GSDMD and NINJ1 is not required to support our conclusion that NINJ1-dependent membrane rupture is involved in inflammasome-pyroptosis induced coagulation and inflammation.

- What are the levels of plasma TAT, PT, and inflammatory cytokines if you collect plasma after 90 minutes? Given the majority (~70%) of the ninj+/- mice are dead by 24 hours, it is imperative to determine whether the 90-minute timeframe data (in Fig 1A-G) is also representative of later time points. The question is whether ninj1+/- just delays the increases in prothrombin time, plasma TAT, and plasma cytokines.

Thank for the valuable question. The time point (90 min) was chosen based on our *in vitro* observation that flagellin-induced pyroptosis in BMDMs largely occurs within 60-90 min.

Because our focus on the primary effect of flagellin *in vivo*, potential secondary effects at later points may complicate the results and are hard to interpret. As the reviewer suggested, we have measured plasma PT, TAT at 6 hours post-flagellin challenge. The significant difference in PT sustained between Ninj1+/+ and Ninj1+/- (Fig A), suggesting coagulation proteins remained more depleted in Ninj1+/+ mice than in Ninj1+/- mice. However, plasma TAT levels were diminished to baseline level (refer to Fig 1B in main text) in both groups and showed no significant difference between groups (Fig B), which could be explained by the short half-life (less than 30 min) in the blood. Since flagellin challenge is a one-time hit, there might not a second episode of coagulation after the 90-minute time point, at least not triggered by flagellin, supported by the plasma TAT levels at 6 hours. We now comment on this limitation at the end of the main text.

Based on our previous studies, plasma IL-1β and TNFα peaked at early time point and diminished over time, but plasma IL-6 levels maintained. As shown below, plasma IL-6 appeared higher in Ninj1+/+ compared with Ninj1+/-, but not statistically significant (partly because one missing sample, n = 4 not 5, in Ninj1+/+ group decreased the statistical power of detecting a difference).

**Author response image 1. sa3fig1:** Mice were injected with Fla (500 ng lFn-Fla plug3 ugPA). Blood was collected 6 hours after Fla injection. Prothrombin time (**A**), plasma TAT (**B**), and plasma IL-6 (**C**) were measured. Mann-Whitney test were performed.

**Recommendations for the authors:**

**Reviewer #1 (Recommendations For The Authors):**
- Fig 1F: are there lower magnification images that capture the fibrin deposition? The IHC data seems at odds with the WB data in Fig. 1G where there is still significant fibrin detected in the heterozygous lungs and liver. Quantitating the Fig. 1G Western blot would also be helpful.

IHC surveys a thin layer of tissue section while WB surveys a piece of tissue, therefore fibrin deposition may be missing from IHC and but found in WB. That is why we used two methods. Below we provide lower mag images of fibrin deposition (about 2 x 1.6 mm area).

**Author response image 2. sa3fig2:** 

- Fig1H - lethality study uses 5x dose of Fla used in earlier studies. In the lethality data where there is a delay in ninj1+/- mortality, are the parameters (prothrombin time, plasma TAT, and plasma cytokines) measured at 90 minutes different between WT and ninj+/- mice? This would be critical to confirm that this is not merely due to a delayed release of TF-containing MVs.

We used 5x lower dose of Fla in coagulation study than lethality study because it’s not as easy to draw blood from septic mouse with higher dose of flagellin. We need to terminate the mice to collect blood for plasma measurement and therefore the parameters were not measured for mice in lethality study.

- What is the effect of ninj+/- on *E. coli*-induced lethality in mice? How do these data compare to *E. coli* infection of GSDMD-/- mice?

We did not examine the effect of Ninj1+/- on *E. coli*-induced lethality. After the initial submission of our manuscript, we have focused on Ninj1 flox/flox mice instead of Ninj1+/- for NINJ1 deficiency. We are using induced global Ninj1 deficient mice for polymicrobial infectioninduced lethality in our new studies.

- Fig 2 - in the *E. coli* model, the prothrombin time, plasma TAT, and plasma cytokines are measured 6 hours post-infection. How were these time points chosen? Did the authors measure prothrombin time, plasma TAT, and plasma cytokines at different time points?

The *in vivo* time point for flagellin and *E. coli* were chosen based on our *in vitro* observation of the timelines on BMDM pyroptosis induced by flagellin and bacteria. This disparity probably arises from distinct dynamics between purified protein and bacterial infections. Purified proteins can swiftly translocate into cells and take effect immediately after injection. Conversely, during bacterial infection, macrophages engulf and digest the bacteria to expose their antigens. Subsequently, these antigens initiate further effects, a process that takes some time to unfold.

Our focus is on the primary effect of flagellin *in vivo*, potential secondary effects at later points may complicate the results and are hard to interpret. As the reviewer suggested, we have measured plasma PT, TAT at 6 hours post-flagellin challenge. The significant difference in PT sustained between Ninj1+/+ and Ninj1+/- (Fig A), suggesting coagulation proteins remained more depleted in Ninj1+/+ mice than in Ninj1+/- mice. However, plasma TAT levels were diminished to baseline level (refer to Fig 1B in main text) in both groups and showed no significant difference between groups (Fig B), which could be explained by the short half-life (less than 30 min) in the blood. Since flagellin challenge is a one-time hit, there might not a second episode of coagulation after the 90-minute time point, at least not triggered by flagellin, supported by the plasma TAT levels at 6 hours. We now comment on this limitation at the end of the main text.

Based on our previous studies, plasma IL-1β and TNFα peaked at early time point and diminished over time, but plasma IL-6 levels maintained. As shown below, plasma IL-6 appeared higher in Ninj1+/+ compared with Ninj1+/-, but not statistically significant (partly because one missing sample, n = 4 not 5, in Ninj1+/+ group decreased the statistical power of detecting a difference).

- Fig 3 - the sequence of figure panels listed in the legend needs to be corrected. Fig 3A requires quantitation of NINJ1 levels compared to beta-actin. Fig 3C - needs a control for equal MV loading.

Thanks for the recommendations. The figure sequence has been corrected. There remain no common markers or loading controls for MV, so we use equal plasma volume for loading control.

Additional comments:(1) In Fig 3A, the size of NINJ1 appears to be increased in the NINJ+/- group.

This discrepancy is likely attributed to a technical issue when running the protein gel and protein transfer, which makes the image tilt to one side.

(2) Describe the method of BMDM isolation.

Thanks for the recommendations. We now include the method of BMDM isolation. In brief, mouse femur and tibia from one leg are harvested and rinsed in ice-cold PBS, followed by a brief rinse in 70% ethanol for 10-15 seconds. Both ends of the bones are then cut open, and the bone marrow is flushed out using a 10 ml syringe with a 26-gauge needle. The marrow is passed through a 19-gauge needle once to disperse the cells. After filtering through a 70-μm cell strainer, the cells are collected by centrifugation at 250 g for 5 minutes at 4 °C, then suspended in two 150 mm petri dish, each containing 25 ml of L-cell conditioned medium (RPMI-1640 supplemented with 10% FBS, 2mM L-Glutamine, 10mM HEPES, 15% LCM, and penicillin/streptomycin). After 3 days, 15 mL of LCM medium is added to each dish cells. The cells typically reach full confluency by days 5-7.

(3) According to this method, BMDMs are seeded without any M-CSF or L929-cell conditioned medium. How many macrophages survive under this condition?

BMDMs are cultured and differentiated in medium supplemented with 15% L929-cell conditioned medium. For the experiment, the cells were seeded in Opti-MEM medium (Thermo Fisher Scientific, Cat# 51985034) without M-CSF or L929-cell conditioned medium. BMDMs can survive under this condition, as evidenced by low LDH and high ATP measurement (Fig S5).

**Reviewer #2 (Recommendations For The Authors):**
- There is significant information missing in the methods and this makes it unclear how to interpret how some of the experiments were performed. For example, there is no detailed description or references in the methods on how the in vivo experiments were performed. The methods section needs significantly more details so that any reader is able to follow the protocols in this manuscript. References to previous work should also be included as needed.

Thanks for the recommendations. We had some of the details in the figure legend. We now add details in the methods for better interpretation of our data.

- Line numbers in the manuscript would be helpful when resubmitting the manuscript so that the reviewer can easily point to the main text when making comments.

Thanks for the recommendations. We now add line numbers in the manuscript.